# Reinforcement Learning with Delayed, Composite, and Partially Anonymous Reward

**Washim Uddin Mondal**                                    *wmondal@purdue.edu*
*School of IE and CE, Purdue University*

**Vaneet Aggarwal**                                         *vaneet@purdue.edu*
*School of IE and ECE, Purdue University*

**Reviewed on OpenReview:** *https://openreview.net/forum?id=ubCoTAynPp*

## Abstract

We investigate an infinite-horizon average reward Markov Decision Process (MDP) with delayed, composite, and partially anonymous reward feedback. The delay and compositeness of rewards mean that rewards generated as a result of taking an action at a given state are fragmented into different components, and they are sequentially realized at delayed time instances. The partial anonymity attribute implies that a learner, for each state, only observes the aggregate of past reward components generated as a result of different actions taken at that state, but realized at the observation instance. We propose an algorithm named DUCRL2 to obtain a near-optimal policy for this setting and show that it achieves a regret bound of $\tilde{\mathcal{O}}\left(DS\sqrt{AT} + d(SA)^3\right)$ where $S$ and $A$ are the sizes of the state and action spaces, respectively, $D$ is the diameter of the MDP, $d$ is a parameter upper bounded by the maximum reward delay, and $T$ denotes the time horizon. This demonstrates the optimality of the bound in the order of $T$, and an additive impact of the delay.

## 1 Introduction

Reinforcement learning (RL) enables an agent to learn a *policy* in an unknown environment by repeatedly interacting with it. The environment has a state that changes as a result of the actions executed by the agent. In this setup, a policy is a collection of rules that guides the agent to take action based on the observed state of the environment. Several algorithms exist in the literature that learn a policy with near-optimal regret (Jaksch et al., 2010; Azar et al., 2017; Jin et al., 2018; Fruit et al., 2018). However, all of the above frameworks assume that the reward is instantly fed back to the learner after executing an action. Unfortunately, this assumption of instantaneity may not hold in many practical scenarios. For instance, in the online advertisement industry, a customer may purchase a product several days after seeing the advertisement. In medical trials, the effect of a medicine may take several hours to manifest. In road networks, it might take a few minutes to notice the impact of a traffic light change. In many of these cases, the effect of an action is not entirely realized in a single instant, but rather fragmented into smaller components that are sequentially materialized over a long interval. Reward feedback that satisfies this property is referred to as delayed and composite reward. In several applications, the learner cannot directly observe each delayed component of a composite reward but only an aggregate of the components realized at the observation instance. For example, in the case of multiple advertisements, the advertiser can only see the total number of purchases at a given time but is completely unaware of which advertisement resulted in what fraction of the total purchase. Such feedback is referred to as anonymous reward.

Learning with delayed, composite, and anonymous rewards is gaining popularity in the RL community. While most of the theoretical analysis has been directed towards the multi-arm bandit (MAB) framework (Wang et al., 2021; Zhou et al., 2019; Vernade et al., 2020; Pike-Burke et al., 2018; Pedramfar and Aggarwal, 2023), recent studies have also analyzed Markov Decision Processes (MDPs) with delayed feedback (Howson

et al., 2023; Jin et al., 2022; Lancewicki et al., 2022). However, none of these studies have incorporated both composite and anonymous rewards, which is the focus of this paper.

## 1.1 The Challenge and Our Contribution

Learning with delayed, composite, and anonymous rewards has gained popularity in the RL community. While there has been extensive theoretical analysis of these types of rewards in the multi-arm bandit (MAB) framework, extending these ideas to the full Markov decision process (MDP) setting poses significant challenges. For example, one of the main ideas used in the MAB framework can be stated as follows (Wang et al., 2021). The learning algorithm proceeds in multiple epochs with the sizes of the epochs increasing exponentially. At the beginning of each epoch, an action is chosen and it is applied at all instances within that epoch. Due to multiple occurrences of the chosen action, and exponentially increasing epoch lengths, the learner progressively obtains better estimates of the associated rewards despite the composite and anonymous nature of the feedback. In an MDP setting, however, it is impossible to ensure that any state-action pair $(s, a)$ appears contiguously over a given stretch of time. The most that one can hope to ensure is that each state, $s$, when it appears in an epoch (defined appropriately), is always paired with a unique action, $a$. In this way, if the state in consideration is visited sufficiently frequently in that epoch, the learner would obtain an accurate estimate of the reward associated with the pair $(s, a)$. Unfortunately, in general, there is no way to ensure a high enough visitation frequency for all states. Unlike the MAB setup, it is, therefore, unclear how to develop regret optimal learning algorithms for MDPs with delayed, composite, and anonymous rewards.

Our article provides a partial resolution to this problem. In particular, we demonstrate that if the rewards are delayed, composite, and *partially anonymous*, then an algorithm can be designed to achieve near-optimal regret. In a fully anonymous setting, a learner observes the aggregate of the (delayed) reward components generated as a result of all state-action pairs visited in the past. In contrast, a partially anonymous setup allows a learner to observe the sum of (delayed) reward components generated as a result of all past visitations to any specified state. Our proposed algorithm, DUCRL2, is built upon the UCRL2 algorithm of (Jaksch et al., 2010) and works in multiple epochs. Unlike the bandit setting, however, the epoch lengths are not guaranteed to be exponentially increasing. Our primary innovation lies in demonstrating how an accurate reward function estimate can be obtained using the partially anonymous feedback. DUCRL2 yields a regret bound of $\tilde{\mathcal{O}}(DS\sqrt{AT} + d(SA)^3)$ where $S, A$ denote the sizes of the state and action spaces respectively, $T$ is the time horizon, $D$ denotes the diameter of the MDP, and the parameter $d$ is bounded by the maximum delay in the reward generation process. The obtained result matches a well-known lower bound in $T$.

## 1.2 Relevant Literature

Below we describe in detail the relevant literature.

**Regret Bounds in Non-delayed RL:** The framework of regret minimization in RL with immediate feedback is well-investigated in the literature. In particular, this topic has been explored in the settings of both stochastic (Jaksch et al., 2010; Zanette et al., 2020; Agarwal and Aggarwal, 2023; Agarwal et al., 2022; 2023) and adversarial (Jin and Luo, 2020; Rosenberg and Mansour, 2019; Shani et al., 2020) MDPs. Our setup can be considered to be the generalization of the stochastic MDPs with immediate feedback.

**Delay in Bandit:** Delayed feedback is a well-researched topic in the bandit literature, with numerous studies conducted in both stochastic (Vernade et al., 2020; Pike-Burke et al., 2018; Zhou et al., 2019; Gael et al., 2020; Lancewicki et al., 2021; Pedramfar and Aggarwal, 2023) and adversarial settings (Quanrud and Khashabi, 2015; Cesa-Bianchi et al., 2016; Thune et al., 2019; Zimmert and Seldin, 2020; Bistritz et al., 2019; Ito et al., 2020). However, as previously discussed, applying the insights of bandit learning to the MDP setting with composite and anonymous rewards is challenging.

**Delay in MDP:** A number of recent papers have explored the incorporation of delayed feedback into the MDP framework. For example, (Lancewicki et al., 2022; Jin et al., 2022) consider adversarial MDPs, while (Howson et al., 2023) analyzes a stochastic setting. However, all of these articles focus on episodic MDPs with non-composite and non-anonymous rewards, which is distinct from our work on infinite-horizon MDPs with delayed, composite, and partially anonymous rewards. It is worth noting that while delayed reward is a

commonly studied topic in the literature, some works also consider delays in the state information (Agarwal and Aggarwal, 2021; Bouteiller et al., 2021). Additionally, the impact of delay has also been explored in the context of multi-agent learning to characterize coarse correlated equilibrium (Zhang et al., 2022).

## 2 Problem Setting

We consider an infinite-horizon average-reward Markov Decision Process (MDP) defined as, $M \triangleq \{\mathcal{S}, \mathcal{A}, r, p\}$ where $\mathcal{S}, \mathcal{A}$ denote the state, and action spaces respectively, $r : \mathcal{S} \times \mathcal{A} \to [0, 1]$ is the reward function, and $p : \mathcal{S} \times \mathcal{A} \to \Delta(\mathcal{S})$ indicates the state transition function. The function, $\Delta(\cdot)$ defines the probability simplex over its argument set. The cardinality of the sets $\mathcal{S}$ and $\mathcal{A}$ are denoted as $S, A$ respectively. Both the reward function, $r$, and the transition function, $p$ are assumed to be unknown.

A learner/agent interacts with an environment governed by the MDP stated above as follows. The interaction proceeds in discrete time steps, $t \in \{1, 2, \cdots\}$. At the time $t$, the state occupied by the environment is denoted as $s_t \in \mathcal{S}$. The learner observes the state, $s_t$, and chooses an action $a_t \in \mathcal{A}$ following a predefined protocol, $\mathbb{A}$. As a result, a sequence of rewards $\boldsymbol{r}_t(s_t, a_t) \triangleq \{r_{t,\tau}(s_t, a_t)\}_{\tau=0}^{\infty}$ is generated where $r_{t,\tau}(s_t, a_t)$ is interpreted as the non-negative component of the vector $\boldsymbol{r}_t(s_t, a_t)$ that is realised at instant $t+\tau$. The following assumption is made regarding the reward generation process.

**Assumption 1** *It is assumed that* $\forall t \in \{1, 2, \cdots\}$, $\forall (s, a) \in \mathcal{S} \times \mathcal{A}$, *the following holds.*

$$(a) \ ||\boldsymbol{r}_t(s, a)||_1 \sim D(s, a) \in \Delta[0, 1],$$
$$(b) \ \mathbb{E}||\boldsymbol{r}_t(s, a)||_1 = r(s, a) \in [0, 1],$$
$$(c) \ \{\boldsymbol{r}_t(s, a)\}_{t \geq 1, (s,a) \in \mathcal{S} \times \mathcal{A}} \ \text{are mutually independent},$$

*where* $|| \cdot ||_1$ *denotes the* 1*-norm.*

Assumption 1(a) dictates that the reward sequences generated from $(s, a)$ are such that the sum of their components can be thought of as samples taken from a certain distribution, $D(s, a)$, over $[0, 1]$. Note that the distribution, $D(s, a)$, is independent of $t$. Assumption 1(b) explains that the expected value of the sum of the reward components generated from $(s, a)$ equals $r(s, a) \in [0, 1]$. Finally, Assumption 1(c) clarifies that the reward sequences generated at different instances (either by the same or distinct state-action pairs) are presumed to be independent of each other.

At the time instant $t$, the learner observes a reward vector $\boldsymbol{x}_t \in \mathbb{R}^S$ whose $s$-th element is expressed as follows.

$$\boldsymbol{x}_t(s) = \sum_{0 < \tau \leq t} r_{\tau, t-\tau}(s, a_\tau) 1(s_\tau = s)$$

where $1(\cdot)$ defines the indicator function. Note that the learner only has access to the lump-sum reward $\boldsymbol{x}_t(s)$, not its individual components contributed by past actions. This explains why the reward is termed as partially anonymous. In a fully anonymous setting, the learner can only access $||\boldsymbol{x}_t||_1$, not its elements. Although full anonymity might be desirable for many practical scenarios, from a theoretical standpoint, it is notoriously difficult to analyze. We discuss this topic in detail in section 4.2. The expected accumulated reward generated up to time $T$ can be computed as,

$$R(s_1, \mathbb{A}, M, T) = \sum_{t=1}^{T} ||\boldsymbol{r}_t(s_t, a_t)||_1 \tag{1}$$

We would like to emphasize that $R(s_1, \mathbb{A}, M, T)$ is not the sum of the observed rewards up to time $T$. Rather, it equates to the sum of all the reward components that are generated as a consequence of the actions taken up to time $T$. We define the quantity expressed below

$$\rho(s_1, \mathbb{A}, M) \triangleq \lim_{T \to \infty} \frac{1}{T} \mathbb{E}[R(s_1, \mathbb{A}, M, T)] \tag{2}$$

as the average reward of the MDP $M$ for a given protocol $\mathbb{A}$ and an initial state $s_1 \in \mathcal{S}$. Here the expectation is obtained over all possible $T$-length trajectories generated from the initial state $s_1$ following the protocol $\mathbb{A}$ and the randomness associated with the reward generation process for any given state-action pair. It is well known that (Puterman, 2014) there exists a stationary deterministic policy $\pi^* : \mathcal{S} \to \mathcal{A}$ that maximizes the average reward $\forall s_1 \in \mathcal{S}$ if $D(M)$, the diameter of $M$ (defined below) is finite. Also, in that case, $\rho(s_1, \pi^*, M)$ becomes independent of $s_1$ and thus can be simply denoted as $\rho^*(M)$. The diameter $D(M)$ is defined as follows.

$$D(M) \triangleq \max_{s \neq s'} \min_{\pi : \mathcal{S} \to \mathcal{A}} \mathbb{E}\left[T(s'|s, \pi, M)\right]$$

where $T(s'|s, \pi, M)$ denotes the time needed for the MDP $M$ to reach the state $s'$ from the state $s$ following the stationary deterministic policy, $\pi$. Mathematically, $\Pr(T(s'|s, \pi, M) = t) \triangleq \Pr(s_t = s|s_1 = s, s_\tau \neq s, 1 < \tau < t, s_\tau \sim p(s_{\tau-1}, a_{\tau-1}), a_\tau \sim \pi(s_\tau))$. In simple words, given two arbitrary distinct states, one can always find a stationary deterministic policy such that the MDP, $M$, on average, takes at most $D(M)$ time steps to transition from one state to the other.

We define the performance of a protocol, $\mathbb{A}$ by the regret it accumulates over a horizon, $T$ which is mathematically expressed as,

$$\mathrm{Reg}(s_1, \mathbb{A}, M, T) = T\rho^*(M) - R(s_1, \mathbb{A}, M, T) \tag{3}$$

where $\rho^*(M)$ is the maximum of the average reward given in (2), and the second term is defined in (1). We would like to mention that in order to define regret, we use $R(s_1, \mathbb{A}, M, T)$ rather than the expected sum of the *observed* rewards up to time $T$. The rationale behind this definition is that all the components of the rewards that are generated as a consequence of the actions taken up to time $T$ would eventually be realized if we allow the system to evolve for a long enough time. Our goal in this article is to come up with an algorithm that achieves sublinear regret for the delayed, composite, and anonymous reward MDP described above.

Before concluding, we would like to provide an example of an MDP with partially anonymous rewards. Let us consider the $T$ round of interaction of a potential consumer with a website that advertises $S$ categories of products. At round $t$, the state, $s_t \in \{1, \cdots, S\}$ observed by the website is the category of product searched by the consumer. The $s$-th category where $s \in \{1, \cdots, S\}$ has $N_s$ number of potential advertisements, and the total number of advertisements is $N = \sum_s N_s$. The website, in response to the observed state, $s_t$, shows an ordered list of $K < N$ advertisements (denoted by $a_t$), some of which may not directly correspond to the searched category, $s_t$. This may cause the consumer, with some probability, to switch to a new state, $s_{t+1}$ in the next round. For example, if the consumer is searching for "Computers", then showing advertisements related to "Mouse" or "Keyboard" may incentivize the consumer to search for those categories of products. After $\tau$ rounds, $0 < \tau \leq T$, the consumer ends up spending $r_\tau(s)$ amount of money for the $s$-th category of product as a consequence of previous advertisements. Note that if the same state $s$ appears in two different rounds $t_1 < t_2 < \tau$, the website can potentially show two different ordered lists of advertisements, $a_{t_1}, a_{t_2}$ to the consumer. However, it is impossible to segregate the portions of the reward $r_\tau(s)$ contributed by each of those actions. Hence, the system described above can be modeled by an MDP with delayed, composite, and partially anonymous rewards.

## 3 DUCRL2 Algorithm

In this section, we develop Delayed UCRL2 (DUCRL2) algorithm to achieve the overarching goal of our paper. It is inspired by the UCRL2 algorithm suggested by (Jaksch et al., 2010) for MDPs with immediate rewards (no delay). Before going into the details of the DUCRL2 algorithm, we would like to introduce the following assumption.

**Assumption 2** *There exists a finite positive number $d$ such that, $\forall t \in \{1, 2 \cdots\}$,*

$$\sum_{\tau_1 \geq 0} \max_{(s,a) \in \mathcal{S} \times \mathcal{A}} \left[\sum_{\tau \geq \tau_1} r_{t,\tau}(s, a)\right] \leq d \tag{4}$$

---

**Algorithm 1** DUCRL2 Algorithm

---
1: **Input:** $\delta \in (0,1)$, $d$, $\mathcal{S}$, $\mathcal{A}$
2: **Initialization:** Observe the initial state $s_1 \in \mathcal{S}$ and set $t \leftarrow 1$, $t_0 \leftarrow 0$.

3: **for** episodes $k \in \{1, 2, \cdots\}$ **do**                                        ▷ Computing empirical estimates
4:     $t_k \leftarrow t$
5:     **for** $(s,a) \in \mathcal{S} \times \mathcal{A}$ **do**
6:         $\nu_k(s,a) \leftarrow 0$
7:         $\mathcal{E}_j(s,a) \leftarrow 1\left((s,a) \in \{(s_\tau, a_\tau) | t_{j-1} \leq \tau < t_j\}\right)$, $\forall j \in \{1, \cdots, k-1\}$
8:         $E_k(s,a) \leftarrow \sum_{0<j<k} \mathcal{E}_j(s,a)$
9:         $N_k(s,a) \leftarrow \sum_{0<\tau<t_k} 1(s_\tau = s, a_\tau = a)$
10:        $\hat{r}_k(s,a) \leftarrow \sum_{0<j<k} \sum_{t_j \leq \tau < t_{j+1}} \boldsymbol{x}_\tau(s) \mathcal{E}_j(s,a) / \max\{1, N_k(s,a)\}$

11:        **for** $s' \in \mathcal{S}$ **do**
12:            $\hat{p}_k(s'|s,a) = \sum_{0<\tau<t_k} 1(s_\tau = s, a_\tau = a, s_{\tau+1} = s') / \max\{1, N_k(s,a)\}$
13:        **end for**
14:    **end for**
15: **end for**

16: Let $\mathcal{M}_k$ be the set of MDPs with state space $\mathcal{S}$, action space $\mathcal{A}$, transition probability $\tilde{p}$, and reward function $\tilde{r}$ such that

$$|\tilde{r}(s,a) - \hat{r}_k(s,a)| \leq \sqrt{\frac{7\log(2SAt_k/\delta)}{2\max\{N_k(s,a),1\}}} + d\frac{E_k(s,a)}{\max\{N_k(s,a),1\}} \tag{5}$$

$$||\tilde{p}(\cdot|s,a) - \hat{p}_k(\cdot|s,a)||_1 \leq \sqrt{\frac{14S\log(2At_k/\delta)}{\max\{1, N_k(s,a)\}}} \tag{6}$$

17: Using extended value function iteration (Algorithm 2), obtain a stationary deterministic policy $\tilde{\pi}_k$ and an MDP $\tilde{M}_k \in \mathcal{M}_k$ such that,

$$\tilde{\rho}_k \triangleq \min_{s \in \mathcal{S}} \rho(s, \tilde{\pi}_k, \tilde{M}_k) \geq \max_{M' \in \mathcal{M}_k, \pi, s' \in \mathcal{S}} \rho(s', \pi, M') - \frac{1}{\sqrt{t_k}} \tag{7}$$

18: **while** $\nu_k(s_t, \tilde{\pi}_k(s_t)) < \max\{1, N_k(s_t, \tilde{\pi}_k(s_t))\}$ **do**
19:     Execute $a_t = \tilde{\pi}_k(s_t)$
20:     Observe the reward $r_t$ and the next state $s_{t+1}$
21:     $\nu_k(s_t, a_t) \leftarrow \nu_k(s_t, a_t) + 1$
22:     $t \leftarrow t + 1$
23: **end while**

---

*with probability 1 where $\{r_{t,\tau}(s,a)\}_{\tau=0}^\infty$ is the reward sequence generated by $(s,a)$ at time $t$.*

Note that if the maximum delay is $d_{\max}$, then using Assumption 1(a), one can show that $d \leq d_{\max}$. Therefore, $d$ can be thought of as a proxy for the maximum delay. To better understand the intuition behind Assumption 2, consider an interval $\{1, \cdots, T_1\}$. Clearly, the reward sequence generated at $t_1 = T_1 - \tau_1$, $\tau_1 \in \{0, \cdots, T_1 - 1\}$, is $\mathbf{r}_{t_1}(s_{t_1}, a_{t_1})$. The portion of this reward that is realized after $T_1$ is expressed by the following quantity: $\sum_{\tau \geq \tau_1} r_{t_1, \tau}(s_{t_1}, a_{t_1})$ which is upper bounded by $\max_{(s,a)} \sum_{\tau \geq \tau_1} r_{t_1, \tau}(s,a)$. Therefore, the total amount of reward that is generated in $\{1, \cdots, T_1\}$ but realized after $T_1$ is bounded by $\sum_{\tau_1 \geq 0} \max_{(s,a)} \sum_{\tau \geq \tau_1} r_{t_1, \tau}(s,a)$. As the distribution of the reward sequence $\mathbf{r}_{t_1}$ is the same for all $t_1$ (Assumption 1(a)), one can replace the $\tau_1$ dependent term $t_1$ with a generic quantity, $t$. Thus, Assumption 2 essentially states that the *spillover* of the rewards generated within any finite interval $\{1, \cdots, T_1\}$ can be bounded by the term $d$.

We would also like to emphasize that if $d_{\max}$ is infinite, then $d$ may or may not be infinite depending on the reward sequence. For example, consider a deterministic sequence whose components are $r_{t,\tau}(s,a) = 2^{-1-\tau}$, $(s,a) \in \mathcal{S} \times \mathcal{A}$, $\forall t \in \{1, 2, \cdots\}$, $\forall \tau \in \{0, 1, \cdots\}$. It is easy to show that one can choose $d = 2$ though $d_{\max}$ is

infinite. On the other hand, if $r_{t,\tau}(s,a) = \mathbf{1}(\tau = \tau')$, $\forall (s,a)$, $\forall t, \forall \tau$ where $\tau'$ is a random variable with the distribution $\Pr(\tau' = k) = (1-p)p^k$, $\forall k \in \{0,1,\cdots\}$, $0 < p < 1$, then one can show that (4) is violated with probability at least $p^d$. In other words, Assumption 2 is not satisfied for any finite $d$.

The DUCRL2 algorithm (Algorithm 1) proceeds in multiple epochs. At the beginning of epoch $k$, i.e., at time instant $t = t_k$, we compute two measures for all state-action pairs. The first measure is indicated as $N_k(s,a)$ which counts the number of times the pair $(s,a)$ appears before the onset of the $k$th epoch. The second measure is $\mathcal{E}_j(s,a)$ which is a binary random variable that indicates whether the pair $(s,a)$ appears at least once in the $j$th epoch, $0 < j < k$. Due to the very nature of our algorithm (elaborated later), in a given epoch $j$, if $\mathcal{E}_j(s,a) = 1$, then $\mathcal{E}_j(s,a') = 0$, $\forall a' \in \mathcal{A} \setminus \{a\}$. Taking a sum over $\{\mathcal{E}_j(s,a)\}_{0<j<k}$, we obtain $E_k(s,a)$ which counts the number of epochs where $(s,a)$ appears at least once before $t_k$, the start of the $k$th episode.

Next, we obtain the reward estimate $\hat{r}_k(s,a)$ by computing the sum of the $s$-th element of the observed reward over all epochs where $(s,a)$ appears at least once and dividing it by $N_k(s,a)$. Also, the transition probability estimate $\hat{p}_k(s'|s,a)$ is calculated by taking a ratio of the number of times the transition $(s,a,s')$ occurs before the $k$th episode and $N_k(s,a)$. It is to be clarified that, due to the delayed composite nature of the reward, the observed reward values that are used for computing $\hat{r}_k(s,a)$ may be contaminated by the components generated by previous actions which could potentially be different from $a$. Consequently, it might appear that the empirical estimates $\hat{r}_k(s,a)$ may not serve as a good proxy for $r(s,a)$. However, the analysis of our algorithm exhibits that by judiciously steering the exploration, it is still possible to obtain a near-optimal regret.

Using the estimates $\hat{r}_k(s,a)$, $\hat{p}_k(\cdot|s,a)$, we now define a confidence set $\mathcal{M}_k$ of MDPs that is characterized by (5), (6). The confidence radius given in (5) is one of the main differences between our algorithm and the UCRL2 algorithm given by (Jaksch et al., 2010). Applying extended value iteration (Appendix A), we then derive a policy $\tilde{\pi}_k$, and an MDP $\tilde{M}_k \in \mathcal{M}_k$ that are near-optimal within the set $\mathcal{M}_k$ in the sense of (7). We keep on executing the policy $\tilde{\pi}_k$ until for at least one state-action pair $(s,a)$, its total number of occurrences within the current epoch, $\nu_k(s,a)$ becomes at least as large as $N_k(s,a)$, the number of its occurrences before the onset of the current epoch. When this criterion is achieved, a new epoch begins and the process described above starts all over again. Observe that, as the executed policies are deterministic, no two distinct pairs $(s,a), (s,a')$ can appear in the same epoch for a given state, $s$.

We would like to conclude with the remark that all the quantities used in our algorithm can be computed in a recursive manner. Consequently, similar to UCRL2, the space complexity of our algorithm turns out to be $\mathcal{O}(S^2 A)$ which is independent of $T$.

## 4    Regret Analysis

Below we state our main result.

**Theorem 1** *Let $D \triangleq D(M)$. With probability at least $1 - \delta$, $\delta \in (0,1)$, for arbitrary initial state $s$, the regret accumulated by the algorithm* DUCRL2 *over $T > 1$ steps can be bounded above as follows.*

$$\text{Reg}(s, \text{DUCRL2}, M, T) \le 34DS\sqrt{AT \log\left(\frac{T}{\delta}\right)} + 2d(SA)^3 \left[\log_2\left(\frac{8T}{SA}\right)\right]^2 \tag{8}$$

*Substituting $\delta = 1/T$, we can therefore bound the expected regret as,*

$$\mathbb{E}\left[\text{Reg}(s, \text{DUCRL2}, M, T)\right] \le 68DS\sqrt{AT \log(T)} + 2d(SA)^3 \left[\log_2\left(\frac{8T}{SA}\right)\right]^2 \tag{9}$$

Theorem 1 states that the regret accumulated by algorithm DUCRL2 is $\tilde{\mathcal{O}}(DS\sqrt{AT} + d(SA)^3)$ where $\tilde{\mathcal{O}}(\cdot)$ hides the logarithmic factors. (Jaksch et al., 2010) showed that the lower bound on the regret is $\Omega(\sqrt{DSAT})$. As our setup is a generalization of the setup considered in (Jaksch et al., 2010), the same lower bound must

also apply to our model. Moreover, if we consider an MDP instance where the rewards arrive with a constant delay, $d$, then in the first $d$ steps, due to lack of any feedback, each algorithm must obey the regret lower bound $\Omega(d)$. We conclude that the regret lower bound of our setup is $\Omega(\max\{\sqrt{DSAT}, d\}) = \Omega(\sqrt{DSAT} + d)$. Although it matches the orders of $T$, and $d$ of our regret upper bound, there is still room for improvement in the orders of $D$, $S$ and $A$.

### 4.1 Proof Sketch of Theorem 1

In this section, we provide a brief sketch of the proof of Theorem 1.

**Step 1:** The first step is to rewrite the total regret as the sum of regrets accumulated over various epochs. Particularly, we show that with probability at least $1 - \delta/12T^{5/4}$, $\delta > 0$ the following bound holds.

$$\text{Reg}(s, \text{DUCRL2}, M, T) \leq \underbrace{\sum_{k=1}^{m} \text{Reg}_k}_{\triangleq Q_1} + \underbrace{\sqrt{\frac{5T}{8} \log\left(\frac{8T}{\delta}\right)}}_{\triangleq Q_2}$$

The term, $\text{Reg}_k$ can be defined as the regret accumulated over epoch $k$ (a precise definition is given in the appendix), and $m$ is such that $T$ lies in the $(m-1)$th epoch. The additional term, $Q_2$ appears due to the stochasticity of the observed reward instances. We now divide $Q_1$ into two parts as follows.

$$Q_1 = \underbrace{\sum_{k=1}^{m} \text{Reg}_k 1(M \notin \mathcal{M}_k)}_{\triangleq Q_{11}} + \underbrace{\sum_{k=1}^{m} \text{Reg}_k 1(M \in \mathcal{M}_k)}_{\triangleq Q_{12}}$$

**Step 2:** In order to bound $Q_{11}$, it is important to have an estimate of $\Pr(M \notin \mathcal{M}_k)$ which we obtain in Lemma 1. We would like to elaborate that although the bound given in Lemma 1 is similar to that given in (Jaksch et al., 2010), the proof techniques are different. In particular, here we account for the fact that the reward estimates, $\{\hat{r}_k(s, a)\}_{(s,a)\in\mathcal{S}\times\mathcal{A}}$, of the $k$th epoch, are potentially corrupted by delayed effects of past actions. We resolve this problem by proving the following inequality (see (32)).

$$\left| \hat{r}_k(s, a) - \frac{1}{N_k(s, a)} \sum_{0 < \tau < t_k} ||\boldsymbol{r}_\tau(s, a)||_1 1(s_\tau = s, a_\tau = a) \right| \leq d \frac{E_k(s, a)}{\max\{N_k(s, a), 1\}} \tag{10}$$

Here the second term in the LHS denotes an estimate of $r(s, a)$ that the learner would have obtained had there been no delay in the observation of the reward instances. In other words, inequality (10) estimates the gap between an MDP with delayed observations, and a hypothetical MDP without any delayed effects. Observe that the RHS of (10) also appears in the confidence radius (5). Therefore, the cost of incorporating delay is a looser confidence in the reward estimates.

**Step 3:** Using Lemma 1, $Q_{11}$ is bounded by $\sqrt{T}$ with high probability (Lemma 2).

**Step 4:** We now focus on bounding the other term, $Q_{12}$. Lemma 3 shows that $\text{Reg}_k \leq J_k^1 + J_k^2 + J_k^3$ where the precise definition of the terms $\{J_k^i\}_{i\in\{1,2,3\}}$ are given in the appendix B.2. Furthermore, it also proves that the following bound holds with high probability.

$$\sum_{k=1}^{m} J_k^1 1(M \in \mathcal{M}_k) = \tilde{\mathcal{O}}\left( \sqrt{T} + \sum_{k=1}^{m} \sum_{(s,a)\in\mathcal{S}\times\mathcal{A}} \frac{\nu_k(s, a)}{\sqrt{\max\{N_k(s, a), 1\}}} \right)$$

where $\tilde{\mathcal{O}}(\cdot)$ hides logarithmic factors and terms related to $D, S, A$. The other notations are identical to that given in section 3.

**Step 5:** The second term, $J_k^2$ is bounded as follows.

$$J_k^2 \triangleq \sum_{(s,a)\in\mathcal{S}\times\mathcal{A}} \nu_k(s,a)(\tilde{r}_k(s,a) - r(s,a))$$

$$\leq \sum_{(s,a)\in\mathcal{S}\times\mathcal{A}} \nu_k(s,a)|\tilde{r}_k(s,a) - \hat{r}_k(s,a)| + \sum_{(s,a)\in\mathcal{S}\times\mathcal{A}} \nu_k(s,a)|\hat{r}_k(s,a) - r(s,a)|$$

Notice that the first term can be bounded by invoking (5). The same inequality can also be used to bound the second term provided that $M \in \mathcal{M}_k$ which, as we have stated before, is a high probability event (Lemma 1). Using this logic, and some algebraic manipulations, we finally obtain the following high probability bound.

$$\sum_{k=1}^m J_2^k 1(M \in \mathcal{M}_k) = \tilde{\mathcal{O}}\left(\sum_{k=1}^m \sum_{(s,a)\in\mathcal{S}\times\mathcal{A}} \frac{\nu_k(s,a)}{\sqrt{\max\{N_k(s,a),1\}}} + 2dSAm^2\right)$$

**Step 6:** Finally, we obtain the following inequality related to the third term.

$$\sum_{k=1}^m J_k^3 1(M \in \mathcal{M}_k) \leq 2 \sum_{k=1}^m \sum_{(s,a)\in\mathcal{S}\times\mathcal{A}} \frac{\nu_k(s,a)}{\sqrt{\max\{N_k(s,a),1\}}}$$

**Step 7:** Combining, we derive the bound stated below with high probability.

$$Q_1 = Q_{11} + Q_{12} \leq \sqrt{T} + \sum_{i=1}^3 \sum_{k=1}^m J_k^i 1(M \in \mathcal{M}_k)$$

$$= \tilde{\mathcal{O}}\left(\sqrt{T} + \sum_{k=1}^m \sum_{(s,a)\in\mathcal{S}\times\mathcal{A}} \frac{\nu_k(s,a)}{\sqrt{\max\{N_k(s,a),1\}}} + 2dSAm^2\right)$$

**Step 8:** We conclude the proof using Lemma 4 which states that,

$$m \leq SA \log_2\left(\frac{8T}{SA}\right), \text{ and } \sum_{k=1}^m \sum_{(s,a)\in\mathcal{S}\times\mathcal{A}} \frac{\nu_k(s,a)}{\sqrt{\max\{N_k(s,a),1\}}} \leq (\sqrt{2}+1)\sqrt{SAT}$$

### 4.2 Limitation

It is important to understand why our approach works with partial anonymity but not with full anonymity. Fix a state $s \in \mathcal{S}$ and an epoch $j$. Recall from section 3 that no two distinct pairs $(s,a),(s,a')$ can appear in the same epoch. Utilizing this property, we can write down the following relation for a pair $(s,a)$ that appears in the $j$th epoch.

$$\underbrace{\sum_{t_j \leq \tau < t_{j+1}} \boldsymbol{x}_\tau(s)}_{\triangleq R_0} = \underbrace{\sum_{0 < \tau < t_j} \sum_{\tau_1 \geq t_j - \tau} r_{\tau,\tau_1}(s_\tau, a_\tau)1(s_\tau = s)}_{\triangleq R_1} + \underbrace{\sum_{t_j \leq \tau < t_{j+1}} ||\boldsymbol{r}_\tau(s,a)||_1 1(s_\tau = s)}_{\triangleq R_2}$$

$$- \underbrace{\sum_{0 < \tau < t_{j+1}} \sum_{\tau_1 \geq t_{j+1} - \tau} r_{\tau,\tau_1}(s_\tau, a_\tau)1(s_\tau = s)}_{\triangleq R_3} \tag{11}$$

The term, $R_0$ denotes the sum of $s$-th components of all observed reward vectors within epoch $j$. Some portion of $R_0$ is due to actions taken before the onset of the $j$th epoch. This past contribution is denoted by the term, $R_1$. The rest of $R_0$ is entirely contributed by the actions taken within epoch $j$. The term, $R_2$

denotes the sum of all the rewards generated as a result of actions taken within epoch $j$. However, due to the delayed nature of the reward, some portion of $R_2$ will be realized after the $j$th epoch. This spillover part is termed as $R_3$. Using Assumption 2, we can write $0 \leq R_1, R_3 \leq d$ with high probability which leads to the following relation.

$$-d \leq R_0 - R_2 \leq d \tag{12}$$

Eq. (12) is the first step in establishing (10). We would like to emphasize the fact that the term, $R_2$ is entirely contributed by $(s, a)$ pairs appearing in epoch $j$ (i.e., no contamination from actions other than $a$). In other words, although our estimation, $\hat{r}(s, a)$ is based on the contaminated observation $R_0$, we demonstrate that it is not far away from uncontaminated estimations. This key feature makes DUCRL2 successful despite having partial anonymity. On the other hand, if rewards were fully anonymous, then (11) would have changed as follows.

$$\underbrace{\sum_{t_j \leq \tau < t_{j+1}} \sum_{s \in \mathcal{S}} \boldsymbol{x}_\tau(s)}_{\triangleq \tilde{R}_0} = \underbrace{\sum_{0 < \tau < t_j} \sum_{\tau_1 \geq t_j - \tau} r_{\tau, \tau_1}(s_\tau, a_\tau)}_{\triangleq \tilde{R}_1} + \underbrace{\sum_{t_j \leq \tau < t_{j+1}} ||\boldsymbol{r}_\tau(s_\tau, a_\tau)||_1}_{\triangleq \tilde{R}_2} \\ - \underbrace{\sum_{0 < \tau < t_{j+1}} \sum_{\tau_1 \geq t_{j+1} - \tau} r_{\tau, \tau_1}(s_\tau, a_\tau)}_{\triangleq \tilde{R}_3} \tag{13}$$

Note that in (13), the term, $\tilde{R}_2$ is a mixer of contributions from various state-action pairs, unlike $R_2$ in (11). This makes our algorithm ineffective in the presence of full anonymity.

Another limitation of our approach is that the delay parameter $d$ is used as an input to Algorithm 1. One can therefore ask how the regret bound changes if an incorrect estimate, $\hat{d}$ of $d$ is used in the algorithm. One can easily prove that if $\hat{d} > d$, then the regret bound changes to $\mathcal{O}(DS\sqrt{AT} + \hat{d}(SA)^3)$. However, if $\hat{d} < d$, then Lemma 1 no longer works, and consequently, the analysis does not yield any sub-linear regret.

## 5    Conclusion

In this work, we addressed the challenging problem of designing learning algorithms for infinite-horizon Markov Decision Processes with delayed, composite, and partially anonymous rewards. We propose an algorithm that achieves near-optimal performance and derive a regret bound that matches the existing lower bound in the time horizon while demonstrating an additive impact of delay on the regret. Our work is the first to consider partially anonymous rewards in the MDP setting with delayed feedback.

Possible future work includes extending the analysis to the more general scenario of fully anonymous, delayed, and composite rewards, which has important applications in many domains. This extension poses theoretical challenges, and we believe it is an exciting direction for future research. Overall, we hope our work provides a useful contribution to the reinforcement learning community and inspires further research on this important topic.

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

## A  Extended Value Iteration

---
**Algorithm 2** Extended Value Iteration

---
1: **Input:** $\{d(s,a), \mathcal{P}(s,a), \hat{r}(s,a), \mathcal{P}(s,a)\}_{(s,a)\in\mathcal{S}\times\mathcal{A}}$, $\epsilon > 0$
2: **Initialization:** $\boldsymbol{u}_0 \triangleq \{u_0(s)\}_{s\in\mathcal{S}} \leftarrow \boldsymbol{0}$, $i \leftarrow 0$, error $\leftarrow 2\epsilon$
3: **while** error $< \epsilon$ **do**
4:     **for** $s \in \mathcal{S}$ **do**
5:          $u_{i+1}(s) = \max\limits_{a\in\mathcal{A}}\left\{\hat{r}(s,a) + d(s,a) + \max\limits_{p(\cdot)\in\mathcal{P}(s,a)}\sum\limits_{s'\in\mathcal{S}} p(s'|s,a)u_i(s')\right\}$
6:     **end for**
7:     error $\leftarrow \max\limits_{s\in\mathcal{S}}\{u_{i+1}(s) - u_i(s)\} - \min\limits_{s\in\mathcal{S}}\{u_{i+1}(s) - u_i(s)\}$
8:     $i \leftarrow i+1$
9: **end while**

---

Here $d(s,a)$ can be though of as the confidence radius as depicted in (5) whereas $\mathcal{P}(s,a)$ is the set of probability vectors that satisfy (6). Note that the stopping criteria for Algorithm 2 is the following.

$$\max_{s\in\mathcal{S}}\{u_{i+1}(s) - u_i(s)\} - \min_{s\in\mathcal{S}}\{u_{i+1}(s) - u_i(s)\} < \epsilon \tag{14}$$

In the context of Algorithm 1, we can take $\epsilon = 1/\sqrt{t_k}$. Theorem 7 of (Jaksch et al., 2010) guarantees that the greedy policy deduced from the terminal utility vector $\boldsymbol{u}_i$ of Algorithm 2 is $\epsilon$-optimal in the sense of (7) if the set of MDPs whose transition probability distribution $p(\cdot|s,a)$ lies in the confidence set $\mathcal{P}(s,a)$, and the reward function, $r(s,a)$ lies at most $d(s,a)$ distance away from the estimate $\hat{r}(s,a)$, comprises at least one MDP with a finite diameter. As a consequence of this result, we can write the following corollary.

**Corollary 1** *Let, $\mathcal{M}$ be the collection of MDPs whose reward function $r(\cdot,\cdot)$ and transition function, $p(\cdot|\cdot,\cdot)$ satisfy the following for given $\{\hat{r}(s,a), d(s,a), \mathcal{P}(s,a)\}_{(s,a)\in\mathcal{S}\times\mathcal{A}}$.*

$$|r(s,a) - \hat{r}(s,a)| \leq d(s,a),$$
$$p(\cdot|s,a) \in \mathcal{P}(s,a)$$

*If the true MDP, $M$ lies in $\mathcal{M}$, then Algorithm 2 always converges. Moreover, if $\boldsymbol{u}_i$ indicates the terminal utility vector for a given $\epsilon > 0$, and $\forall(s,a) \in \mathcal{S} \times \mathcal{A}$,*

$$(\tilde{\pi}(s), \tilde{p}(\cdot|s,a)) \triangleq \arg\max_{a\in\mathcal{A},\ p(\cdot)\in\mathcal{P}(s,a)}\left\{\hat{r}(s,a) + d(s,a) + \sum_{s'\in\mathcal{S}} p(s'|s,a)u_i(s')\right\}, \tag{15}$$

*then the following inequality holds.*

$$\min_{s\in\mathcal{S}} \rho(s, \tilde{\pi}, \tilde{M}) \geq \max_{M'\in\mathcal{M}, \pi, s'\in\mathcal{S}} \rho(s', \pi, M') - \epsilon$$

*where $\tilde{M}$ is an MDP with transition function, $\tilde{p}(\cdot|\cdot\cdot)$ defined by (15), and reward function $\tilde{r}$ that obeys $\tilde{r}(s,a) = \hat{r}(s,a) + d(s,a)$, $\forall(s,a) \in \mathcal{S} \times \mathcal{A}$.*

Corollary 1 is easily established by observing that the true MDP, $M$ is assumed to have a finite diameter. It is also worthwhile to state that the complexity of updating the vector $\boldsymbol{u}_i$ is $\mathcal{O}(S^2 A)$ as discussed in section 3.1.2 of (Jaksch et al., 2010).

## B  Proof of Theorem 1

Let $T$ be such that $t_{m-1} \leq T < t_m$. Clearly,

$$T < t_m = \sum_{(s,a)\in\mathcal{S}\times\mathcal{A}} N_m(s,a) = \sum_{k=1}^{m} \sum_{(s,a)\in\mathcal{S}\times\mathcal{A}} \nu_k(s,a)$$

Using the above relation, the regret given in (3) can be rewritten as,

$$\text{Reg}(s, \text{DUCRL2}, M, T) = T\rho^*(M) - \sum_{t=1}^{T} ||\boldsymbol{r}_t(s_t, a_t)||_1$$

$$< \sum_{(s,a)\in\mathcal{S}\times\mathcal{A}} N_m(s,a) \left[\rho^*(M) - r(s,a)\right] + \sum_{(s,a)\in\mathcal{S}\times\mathcal{A}} N_m(s,a)r(s,a) - \sum_{t=1}^{T} ||\boldsymbol{r}_t(s_t, a_t)||_1$$

$$= \sum_{k=1}^{m} \underbrace{\sum_{(s,a)\in\mathcal{S}\times\mathcal{A}} \nu_k(s,a) \left[\rho^*(M) - r(s,a)\right]}_{\triangleq \text{Reg}_k} + \sum_{(s,a)\in\mathcal{S}\times\mathcal{A}} N_m(s,a)r(s,a) - \sum_{t=1}^{T} ||\boldsymbol{r}_t(s_t, a_t)||_1$$

The term $\text{Reg}_k$ can be interpreted as the regret accumulated over epoch $k$. Observe that, for a given history of state-action evolution $\mathcal{H}_T \triangleq \{(s_t, a_t)\}_{t=1}^{T}$ up to time $T$, the collection of random variables $\{||\boldsymbol{r}_t(s_t, a_t)||_1\}_{t=1}^{T}$ are mutually independent. Moreover,

$$\mathbb{E}\left[\sum_{t=1}^{T} ||\boldsymbol{r}_t(s_t, a_t)||_1 \Big| \mathcal{H}_T\right] = \sum_{t=1}^{T} r(s_t, a_t) = \sum_{(s,a)\in\mathcal{S}\times\mathcal{A}} r(s,a) \sum_{t=1}^{T} 1(s_t = s, a_t = a)$$

$$= \sum_{(s,a)\in\mathcal{S}\times\mathcal{A}} N_m(s,a)r(s,a)$$

Using Hoeffding 's inequality, we therefore obtain,

$$\Pr\left\{ \sum_{(s,a)\in\mathcal{S}\times\mathcal{A}} N_m(s,a)r(s,a) - \sum_{t=1}^{T} ||\boldsymbol{r}_t(s_t, a_t)||_1 > \sqrt{\frac{5T}{8}\log\left(\frac{8T}{\delta}\right)} \Big| \mathcal{H}_T \right\} \le \frac{\delta}{12T^{\frac{5}{4}}}$$

This implies that, with probability at least $1 - \delta/12T^{\frac{5}{4}}$, the following holds.

$$\text{Reg}(s, \text{DUCRL2}, M, T) \le \sum_{k=1}^{m} \text{Reg}_k + \sqrt{\frac{5T}{8}\log\left(\frac{8T}{\delta}\right)} \tag{16}$$

### B.1 Regret bound on episodes where $M$ lie outside the confidence set

Recall that $\mathcal{M}_k$ is defined to be a collection of MDPs that obey the confidence bounds (5), and (6). In this subsection, we calculate the regret contribution of the episodes where the true MDP $M$ does not satisfy these bounds. The following lemma provides an upper bound estimate of the probability that the true MDP, $M$ does not lie in the confidence set, $\mathcal{M}_k$.

**Lemma 1** $\Pr\{M \notin \mathcal{M}_k\} \le \dfrac{\delta}{15t_k^6}$

The proof of Lemma 1 is relegated to Appendix C. Although the final result in Lemma 1 is the same as in (Jaksch et al., 2010), the proof techniques are quite different. In particular, we need to account for the fact that reward estimates might be corrupted by contributions originating from various past actions. Using Lemma 1, the following bound can be obtained.

**Lemma 2** *(Jaksch et al., 2010) With probability at least $1 - \delta/12T^{\frac{5}{4}}$,*

$$\sum_{k=1}^{m} \text{Reg}_k 1(M \notin \mathcal{M}_k) \le \sqrt{T} \tag{17}$$

We would like to mention here that although the definition of $\mathcal{M}_k$ used in our article is different from that given in (Jaksch et al., 2010), the above result still holds. This is mainly because the only property of $\mathcal{M}_k$ that is invoked to prove Lemma 2 is provided in Lemma 1 which is the same as in (Jaksch et al., 2010).

### B.2 Regret bound on episodes where $M$ lie inside the confidence set

Let $k$ be the index of an episode where $M \in \mathcal{M}_k$ and $\boldsymbol{u}_k = \{u_k(s)\}_{s \in \mathcal{S}}$ be the terminal utility vector obtained via extended value iteration at the $k$th episode. Define, $\boldsymbol{w}_k = \{w_k(s)\}_{s \in \mathcal{S}}$,

$$w_k(s) \triangleq u_k(s) - \frac{\max_{s \in \mathcal{S}} u_k(s) + \min_{s \in \mathcal{S}} u_k(s)}{2}$$

**Lemma 3** *(Jaksch et al., 2010) If $k$ is such that $M \in \mathcal{M}_k$, then,*

$$\mathrm{Reg}_k \leq \underbrace{\sum_{s \in \mathcal{S}} \nu_k(s, \tilde{\pi}_k(s)) \left[ \sum_{s' \in \mathcal{S}} \tilde{p}_k(s'|s, \tilde{\pi}_k(s)) w_k(s') - w_k(s) \right]}_{\triangleq J_k^1}$$

$$+ \underbrace{\sum_{(s,a) \in \mathcal{S} \times \mathcal{A}} \nu_k(s, a)(\tilde{r}_k(s, a) - r(s, a))}_{\triangleq J_k^2} + 2 \underbrace{\sum_{(s,a) \in \mathcal{S} \times \mathcal{A}} \frac{\nu_k(s, a)}{\sqrt{t_k}}}_{\triangleq J_k^3}$$

*where $\tilde{r}_k, \tilde{p}_k, \tilde{\pi}_k$ are the reward, transition function, and policy of the MDP, $\tilde{M}_k$. Moreover,*

$$\sum_{k=1}^m J_k^1 \mathbb{1}(M \in \mathcal{M}_k) \leq D\sqrt{\frac{5}{2} T \log\left(\frac{8T}{\delta}\right)} + DSA \log_2\left(\frac{8T}{SA}\right)$$

$$+ D\sqrt{14S \log\left(\frac{2AT}{\delta}\right)} \sum_{k=1}^m \sum_{(s,a) \in \mathcal{S} \times \mathcal{A}} \frac{\nu_k(s, a)}{\sqrt{\max\{N_k(s, a), 1\}}} \tag{18}$$

*with probability at least $1 - \delta/12T^{\frac{5}{4}}$.*

The only properties of $\mathcal{M}_k$ that are used in the proof of Lemma 3 are (6), and (7) which are the same as given in (Jaksch et al., 2010). Note that, the term $J_k^2$ defined in Lemma 3 can be bounded as follows,

$$J_k^2 \leq \sum_{(s,a) \in \mathcal{S} \times \mathcal{A}} \nu_k(s, a)|\tilde{r}_k(s, a) - \hat{r}_k(s, a)| + \sum_{(s,a) \in \mathcal{S} \times \mathcal{A}} \nu_k(s, a)|\hat{r}_k(s, a) - r(s, a)|$$

$$\overset{(a)}{\leq} \sum_{(s,a) \in \mathcal{S} \times \mathcal{A}} 2\nu_k(s, a)\sqrt{\frac{7 \log\left(\frac{2SAT}{\delta}\right)}{2 \max\{N_k(s, a), 1\}}} + \sum_{(s,a) \in \mathcal{S} \times \mathcal{A}} \frac{2d\nu_k(s, a)}{\max\{N_k(s, a), 1\}} E_k(s, a) \tag{19}$$

where $(a)$ applies the facts that $M, \tilde{M}_k \in \mathcal{M}_k$, and $t_k \leq T$. Note that our algorithm enforces $\nu_k(s, a) \leq \max\{N_k(s, a), 1\}$. Therefore, $J_2^k$ can be further bounded as,

$$J_2^k \leq \sum_{(s,a) \in \mathcal{S} \times \mathcal{A}} \frac{\nu_k(s, a)}{\sqrt{\max\{N_k(s, a), 1\}}} \sqrt{14 \log\left(\frac{2SAT}{\delta}\right)} + 2d \sum_{(s,a) \in \mathcal{S} \times \mathcal{A}} E_k(s, a) \tag{20}$$

Observe that, $E_k(s, a) \leq m$, $\forall (s, a) \in \mathcal{S} \times \mathcal{A}$. Therefore,

$$\sum_{k=1}^m J_2^k \mathbb{1}(M \in \mathcal{M}_k) \leq \sqrt{14 \log\left(\frac{2SAT}{\delta}\right)} \sum_{k=1}^m \sum_{(s,a) \in \mathcal{S} \times \mathcal{A}} \frac{\nu_k(s, a)}{\sqrt{\max\{N_k(s, a), 1\}}} + 2dSAm^2 \tag{21}$$

Finally, injecting the inequality, $\max\{N_k(s, a), 1\} \leq t_k$, we obtain the following bound,

$$\sum_{k=1}^m J_k^3 \mathbb{1}(M \in \mathcal{M}_k) \leq 2 \sum_{k=1}^m \sum_{(s,a) \in \mathcal{S} \times \mathcal{A}} \frac{\nu_k(s, a)}{\sqrt{\max\{N_k(s, a), 1\}}} \tag{22}$$

We now use the following lemma to simplify the upper bounds.

**Lemma 4** *(Jaksch et al., 2010) The following inequalities hold true,*

$$m \leq SA \log_2 \left( \frac{8T}{SA} \right), \tag{23}$$

$$\sum_{k=1}^{m} \sum_{(s,a) \in \mathcal{S} \times \mathcal{A}} \frac{\nu_k(s,a)}{\sqrt{\max\{N_k(s,a), 1\}}} \leq (\sqrt{2} + 1)\sqrt{SAT} \tag{24}$$

Using Lemma 4 and combining (18), (20), (22), we conclude that the following satisfies with probability at least $1 - \delta/12T^{\frac{5}{4}}$.

$$\sum_{k=1}^{m} \mathrm{Reg}_k 1(M \in \mathcal{M}_k) \leq D\sqrt{\frac{5}{2}T \log\left(\frac{8T}{\delta}\right)} + DSA \log_2\left(\frac{8T}{SA}\right)$$

$$+ 2(\sqrt{2}+1)\left[D\sqrt{14S \log\left(\frac{2AT}{\delta}\right)} + 1\right]\sqrt{SAT} + 2d(SA)^3 \left[\log_2\left(\frac{8T}{SA}\right)\right]^2 \tag{25}$$

### B.3 Obtaining the Total Regret

Combining (16), (27), and (25), we can now establish that the following inequality is satisfied with at least $1 - \delta/12T^{5/4} - \delta/12T^{5/4} - \delta/12T^{5/4} = 1 - \delta/4T^{\frac{5}{4}}$ probability.

$$\mathrm{Reg}(s, \mathrm{DUCRL2}, M, T) \leq 34DS\sqrt{AT \log\left(\frac{T}{\delta}\right)} + 2d(SA)^3 \left[\log_2\left(\frac{8T}{SA}\right)\right]^2$$

Taking a union bound on $T$ and noting that $\sum_{T=2}^{\infty} \delta/4T^{\frac{5}{4}} < \delta$, we conclude the theorem.

## C  Proof of Lemma 1

The probability that the $L_1$-deviation between the true and the empirical distributions of $l$ events over $n$ independent sample exceeds $\epsilon$ can be bounded as (Weissman et al., 2003),

$$\Pr\left\{||\hat{p}(\cdot) - p(\cdot)||_1 > \epsilon\right\} \leq (2^l - 2)\exp\left(-\frac{n\epsilon^2}{2}\right) \tag{26}$$

Presume, without loss of generality that, $N_k(s,a) \geq 1$. In our case, inequality (26) can be utilised to obtain the following bound $\forall (s,a) \in \mathcal{S} \times \mathcal{A}$.

$$\Pr\left\{||\hat{p}_k(\cdot|s,a) - p(\cdot|s,a)||_1 > \sqrt{\frac{14S \log\left(2At_k/\delta\right)}{N_k(s,a)}}\right\}$$

$$\overset{(a)}{\leq} \sum_{0 < n < t_k} \Pr\left\{||\hat{p}_k(\cdot|s,a) - p(\cdot|s,a)||_1 > \sqrt{\frac{14S \log\left(2At_k/\delta\right)}{n}}\right\} \tag{27}$$

$$\leq \sum_{0 < n < t_k} 2^S \exp\left(-7S \log\left(2At_k/\delta\right)\right) \leq \sum_{0 < n < t_k} \frac{\delta}{20SAt_k^7} = \frac{\delta}{20SAt_k^6}$$

Inequality $(a)$ is an application of the union bound. Recall that $\mathcal{E}_j(s,a)$ indicates whether the pair $(s,a)$ appears in the $j$th episode. Using this notation, we deduce that, if $\mathcal{E}_j(s,a) = 1$ for some $(s,a) \in \mathcal{S} \times \mathcal{A}$, then,

$$\sum_{t_j \leq \tau < t_{j+1}} \boldsymbol{x}_\tau(s) = \sum_{t_j \leq \tau < t_{j+1}} ||\boldsymbol{r}_\tau(s,a)||_1 1(s_\tau = s) + \sum_{0 < \tau < t_j} \sum_{\tau_1 \geq t_j - \tau} r_{\tau, \tau_1}(s_\tau, a_\tau) 1(s_\tau = s)$$

$$- \sum_{0 < \tau < t_{j+1}} \sum_{\tau_1 \geq t_{j+1} - \tau} r_{\tau, \tau_1}(s_\tau, a_\tau) 1(s_\tau = s) \tag{28}$$

Using Assumption 2, we can now write the following $\forall j$.

$$\sum_{0 < \tau < t_j} \sum_{\tau_1 \geq t_j - \tau} r_{\tau, \tau_1}(s_\tau, a_\tau) 1(s_\tau = s) \leq d \ \ \text{with probability } 1 \tag{29}$$

Combining (28), (29), we can now write the following with probability 1,

$$-d \leq \sum_{t_j \leq \tau < t_{j+1}} \boldsymbol{x}_\tau(s) - \sum_{t_j \leq \tau < t_{j+1}} ||\boldsymbol{r}_\tau(s,a)||_1 1(s_\tau = s) \leq d \tag{30}$$

Taking a sum over all episodes $j \in \{1, \cdots, k-1\}$ where $\mathcal{E}_j(s,a) = 1$, we get the following inequality that is satisfied with probability 1.

$$\left| \sum_{0 < j < k} \sum_{t_j \leq \tau < t_{j+1}} \boldsymbol{x}_\tau(s) \mathcal{E}_j(s,a) - \sum_{0 < \tau < t_k} ||\boldsymbol{r}_\tau(s,a)||_1 1(s_\tau = s, a_\tau = a) \right| \leq d E_k(s,a) \tag{31}$$

Using the definition of $\hat{r}_k(s,a)$, the above inequality can be rephrased as,

$$\left| \hat{r}_k(s,a) - \frac{1}{N_k(s,a)} \sum_{0 < \tau < t_k} ||\boldsymbol{r}_\tau(s,a)||_1 1(s_\tau = s, a_\tau = a) \right| \leq d \frac{E_k(s,a)}{N_k(s,a)} \tag{32}$$

Using Assumption 1(b), we can show that,

$$\mathbb{E}\left[ \frac{1}{N_k(s,a)} \sum_{0 < \tau < t_k} ||\boldsymbol{r}_\tau(s,a)||_1 1(s_\tau = s, a_\tau = a) \right] = r(s,a) \tag{33}$$

Therefore, the following sequence of inequalities can be derived.

$$\Pr \left\{ |\hat{r}_k(s,a) - r(s,a)| > d \frac{E_k(s,a)}{N_k(s,a)} + \sqrt{\frac{7 \log(2SAt_k/\delta)}{2N_k(s,a)}} \right\}$$

$$\overset{(a)}{\leq} \Pr \left\{ \left| \frac{1}{N_k(s,a)} \sum_{0 < \tau < t_k} ||\boldsymbol{r}_\tau(s,a)||_1 1(s_\tau = s, a_\tau = a) - r(s,a) \right| > \sqrt{\frac{7 \log(2SAt_k/\delta)}{2N_k(s,a)}} \right\}$$

$$\overset{(b)}{\leq} \sum_{0 < n < t_k} \Pr \left\{ \left| \frac{1}{n} \sum_{l=1}^{n} ||\boldsymbol{r}_{\tau_l}(s,a)||_1 - r(s,a) \right| > \sqrt{\frac{7 \log(2SAt_k/\delta)}{2n}} \right\}$$

$$\overset{(c)}{\leq} \sum_{0 < n < t_k} \frac{2\delta}{120SAt_k^7} = \frac{\delta}{60SAt_k^6}$$

Inequality $(a)$ is a consequence of (32) while $(b)$ follows from the union bound. Finally, $(c)$ utilizes Hoeffding's inequality together with Assumption 1(a) and 1(c). Conjoining (27) and the above result, we establish the lemma.

