# OpenReview forum: "Reinforcement Learning with Delayed, Composite, and Partially Anonymous Reward"
_TMLR — Accepted by TMLR_

### Review · Reviewer_tkbK · 2023-05-29

**Summary Of Contributions:**

This paper studies the MDPs with delayed and composite rewards. In such a setting, the agent can only observe an aggregate of the historical rewards (on a specific state, in this paper). The author provides a new algorithm based on the UCRL2 for tabular MDPs. The new algorithm consists of a larger confidence bonus regards the delay $d$. The authors provide a sublinear regret bound for averaged MDPs with infinite horizons.

**Audience:**

Yes

**Broader Impact Concerns:**

I do not have any concerns regarding the broader impact of this paper

**Claims And Evidence:**

Yes

**Requested Changes:**

- Please see the aforementioned weakness.
- It will be beneficial to understand the delayed reward setting if the authors can provide some counter-examples regards Assumption 2.
- What if the algorithm has a wrong delay parameter $d$?
- Typos: Right after Assumption 2: $d_m$ or $d_\max$?

**Strengths And Weaknesses:**

I think this paper is an overall interesting paper and provides enough contribution.

#### Strengths

- The paper is well-written and easy to follow. The related works are adequately discussed.
- The assumption of composite, delayed reward is interesting. The algorithm design is intuitive thus it can be extended to related tasks.

#### Weakness

- There lacks some intuition behind Assumption 2 (delayed reward). It's true that a constant delay $d_m$ satisfies this assumption, but it might be better to discuss why this assumption is defined by a double summation.
- I wonder about the space efficiency of the algorithm since the algorithm needs to maintain several composite rewards (e.g. $\bm x_\tau$) in the history. In contrast, In UCRL2, we can reduce the space complexity by only maintaining the summation of the reward $r$.
- The algorithm needs to know the delay $d$, which might be hard to estimate unless the delay is a constant.
- I found the definition of the delayed reward confusing, or maybe the authors missed some definitions of the notations. For example, $r_{t, \tau}$ is defined but $r_\tau$ in Assumption 2 is not defined, should it be something like $r_{0, \tau}$ or do I miss some definitions?

---

> ### Author Response · Authors · 2023-05-31
> **Response**
>
> Thank you for your constructive comments. Please see our response to your comments below.
>
> $\textbf{Q1:}$ Intuition about Assumption 2.
>
> $\textbf{Response:}$ We have added the following passage in the revised manuscript (Section 3) that elaborates the intuition behind assumption 2.
>
> > To better understand the intuition behind Assumption 2, consider an interval $\\{1, \cdots, T_1\\}$. Clearly, the reward sequence generated at  $t_1=T_1-\tau_1$, $\tau_1\in\\{0, \cdots, T_1-1\\}$, is $\mathbf{r}\_{t_1}(s\_{t_1}, a_{t_1})$. The portion of this reward that is realized after $T_1$ is expressed by the following quantity: $\sum\_{\tau\geq \tau_1}r_{t_1,\tau}(s_{t_1}, a_{t_1})$ which is upper bounded by $\max\_{(s, a)} \sum\_{\tau\geq \tau_1} r\_{t_1, \tau}(s, a)$. Therefore, the total amount of reward that is generated in $\\{1, \cdots, T_1\\}$ but realized after $T_1$ is bounded by $\sum\_{\tau_1\geq 0}\max\_{(s, a)} \sum\_{\tau\geq \tau_1} r\_{t_1, \tau}(s, a)$. As the distribution of the reward sequence $\mathbf{r}_{t_1}$ is the same for all $t_1$ (Assumption 1(a)), one can replace the $\tau_1$ dependent term $t_1$ with a generic quantity, $t$. Thus, Assumption 2 essentially states that the $\textit{spillover}$ of the rewards generated within any finite interval $\\{1, \cdots, T_1\\}$ can be bounded by the term $d$.
>
> $\textbf{Q2:}$ On the space efficiency of the algorithm.
>
> $\textbf{Response:}$ The space complexity of our algorithm is $\mathcal{O}(S^2A)$ and it is not dependent on $T$. This is because each term used in the algorithm can be calculated in a recursive manner and storing all historical information is not required. The following text has been incorporated in the revised paper to emphasize this point (Section 3).
>
> > We would like to conclude with the remark that all the quantities used in our algorithm can be computed in a recursive manner. Consequently, similar to UCRL2, the space complexity of our algorithm turns out to be $\mathcal{O}(S^2A)$ which is independent of $T$.
>
> $\textbf{Q3:}$ On the impact of delay estimation.
>
> $\textbf{Response:}$ We have added the following passage in the revised manuscript (Section 4.2).
>
> > Another limitation of our approach is that the delay parameter $d$ is used as an input to Algorithm 1. One can therefore ask how the regret bound changes if an incorrect estimate, $\hat{d}$ of $d$ is used in the algorithm. One can easily prove that if $\hat{d}> d$, then the regret bound changes to $\tilde{\mathcal{O}}(DS\sqrt{AT}+ \hat{d}(SA)^3)$. However, if $\hat{d}<d$, then Lemma 1 no longer works, and consequently, the analysis does not yield any sub-linear regret.
>
> $\textbf{Q4:}$ Clarifying the definition of delayed reward.
>
> $\textbf{Response:}$ In the modified paper, we have changed the notation of $r_{\tau}(s, a)$ to $r_{t, \tau}(s, a)$ to make the notation consistent.
>
> $\textbf{Q5:}$ Counter-examples regarding Assumption 2.
>
> $\textbf{Response:}$ The following counter-example has been provided in the revised paper (Section 3).
>
> > We would also like to emphasize that if $d_{\max}$ is infinite, then $d$ may or may not be infinite depending on the reward sequence. For example, consider a deterministic sequence whose components are  $r\_{t, \tau}(s, a) = 2^{-1-\tau}$, $(s, a)\in \mathcal{S} \times \mathcal{A}$, $\forall t\in \\{1, 2, \cdots\\}$, $\forall \tau\in \\{0, 1, \cdots\\}$. It is easy to show that one can choose $d=2$ though $d_{\max}$ is infinite. On the other hand, if $r\_{t, \tau}(s, a) = \mathbf{1}(\tau = \tau')$, $\forall(s, a)$, $\forall t, \forall\tau$ where $\tau'$ is a random variable with the distribution $\mathrm{Pr}(\tau'=k) = (1-p)p^k$, $\forall k \in \\{0, 1, \cdots\\}$, $0<p<1$, then one can show that  (4) is violated with probability at least $p^d$. In other words, Assumption 2 is not satisfied for any finite $d$.
>
> $\textbf{Q6:}$ Typo regarding the maximum delay.
>
> $\textbf{Response:}$ Thank you for pointing out the typo. In the revised article, we have changed $d_m$ to $d_{\max}$ to make the notation consistent.

---

### Review · Reviewer_2tkA · 2023-05-31

**Summary Of Contributions:**

This work studies the problem of tabular reinforcement learning with delayed feedback. In particular, it considers the infinite-horizon setting, and “partially anonymous” feedback (where the learner can observe the total reward generated by any given state, but is not told which visitations to the state resulted in this reward). They propose a variant of the UCRL2 algorithm to handle this feedback, and prove a bound on its regret with leading-order term scaling similarly to the standard UCRL2 algorithm, and lower-order term scaling with the maximum delay.

**Audience:**

Yes

**Broader Impact Concerns:**

None.

**Claims And Evidence:**

Yes

**Requested Changes:**

Critical requested changes:
- I believe it is important to provide a lower bound showing that linear scaling in $d$ is unavoidable. I suspect this should not be too hard to show (if the learner has no feedback after the first $d$ steps, they will be unable to learn anything, and so it should be straightforward to construct a family of instances for which any algorithm must suffer linear regret until $T > d$ on some instance).
- I believe it is critical that the algorithm be extended to handle unknown $d$. Without this, as noted, the algorithm and result are almost immediate from the analysis given in [Jaksh et al., 2010]. Existing work on the problem of delayed feedback (for example, [Howson et al., 2023]) propose algorithms that do not require knowledge of the delay, so I believe it should be possible. If it is not possible, some discussion on why it is not possible should be given.
- The related work section is very short and should be expanded. In particular, it should include some references covering the tabular RL literature more broadly.
- Some discussion should be given justifying the partially anonymous feedback setting. In the examples used to motivate delayed feedback, is it reasonable to assume partially anonymous feedback is available?
- In addition, while it is argued that partially anonymous feedback is necessary for the proposed algorithm, it is not argued that partially anonymous feedback is necessary for any algorithm. Some discussion of this should be provided.

Important but less critical requested changes:
- The definition of $d$ is somewhat hard to parse and it is not clear where it comes from. I would suggest either adding additional discussion interpreting this quantity, or just stating all results in terms of $d_{\max}$, and putting the result in terms of $d$ in the appendix.
- $T(s’ | s,\pi,M)$, used in the definition of $D(M)$, should be precisely defined.
- I believe the references should go before the appendix.

Several small notational things and typos:
- Is there a reason that “$| \cdot |_1$” is used for the $\ell_1$ norm and not $|| \cdot ||_1$, as is standard in the literature? Similarly, is there a reason that $\Delta$ is used for regret instead of, e.g. $\mathcal{R}$ or $\mathrm{Reg}$? Neither of these are major issues and could be left as is, but it might improve readability if more standard notation is used.
- Assumption 2 uses different notation for the reward than is used in the preliminary section (e.g. $r_{\tau}(s,a)$ instead of $r_{t,\tau}(s,a)$). It would be helpful if these could be made consistent.
- After Assumption 2, I believe “the maximum delay is $d_m$” should be “the maximum delay is $d_{\max}$”. Later in this same paragraph, “though” should be “thought”.
- The line before equation (11) should end in a “:” not a “.”.

**Strengths And Weaknesses:**

Strengths:
- The problem of delayed feedback is interesting and well-motivated, yet has received relatively little attention in the RL community.
- The upper bound scales sublinearly in $T$ and linearly in the length of the delay. The former is known to be tight (although this work inherits suboptimal dependence on other parameters in the leading order term from [Jaksh et al., 2010]), and the scaling in delay is also likely not improvable.

Weaknesses:
- No lower bound is given showing the dependence on the delay is tight. While it does seem intuitively like the linear scaling is necessary, ideally this should be shown formally.
- The algorithm requires knowledge of the maximum delay (or an upper bound on it) and no discussion is given as to how one should remove this assumption.
- The analysis is very standard. The majority of it follows from [Jaksh et al., 2010], with the only modification being to handle the delayed feedback in bounding the reward estimation error. However, given that the algorithm has knowledge of the maximum delay, it is relatively straightforward to bound this.
- While the problem of delayed feedback is well-motivated, it is not clear how well-motivated the model of feedback considered (partially anonymous feedback), and the feedback model feels rather strong.

---

> ### Author Response · Authors · 2023-06-28
> **Response: Part 1**
>
> $\textbf{Q1: On the lower bound of regret.}$
>
> $\textbf{Response:}$ Thank you for pointing out that our suggested regret is optimal both in terms of the horizon, $T$, and the maximum delay, $d_{\max}$. We have added the following paragraph in section 4 of the revised paper.
>
> > (Jacks et. al., 2010) showed that the lower bound on the regret is $\Omega(\sqrt{DSAT})$. As our setup is a generalization of the setup considered in (Jacks et. al., 2010), the same lower bound must also apply to our model. Moreover, if we consider an MDP instance where the rewards arrive with a constant delay, $d$, then in the first $d$ steps, due to lack of any feedback, each algorithm must obey the regret lower bound $\Omega(d)$. We conclude that the regret lower bound of our setup is $\Omega(\max\\{\sqrt{DSAT}, d\\})=\Omega(\sqrt{DSAT}+d)$. Although it matches the orders of $T$, $A$, and $d$ of our regret upper bound, there is still room for improvement in the orders of $D$ and $S$.
>
> $\textbf{Q2: On the knowledge of maximum delay.}$
>
> $\textbf{Response:}$ We agree that the knowledge of the delay parameter, $d$ is indeed required in the algorithm. However, if we apply an upper bound $\hat{d}$ of $d$, instead of its actual value, then the algorithm still works. The pitfall is that, a poor estimation, $\hat{d}$ leads to a larger regret value of $\mathcal{O}(DS\sqrt{AT}+\hat{d}(SA)^3)$. We have added the following passage in the revised manuscript (section 4.2).
>
> > Another limitation of our approach is that the delay parameter $d$ is used as an input to Algorithm 1. One can therefore ask how the regret bound changes if an incorrect estimate, $\hat{d}$ of $d$ is used in the algorithm. One can easily prove that if $\hat{d}> d$, then the regret bound changes to $\mathcal{O}(DS\sqrt{AT}+ \hat{d}(SA)^3)$. However, if $\hat{d}<d$, then Lemma 1 no longer works, and consequently, the analysis does not yield any sub-linear regret.
>
> We thank the reviewer for pointing out that (Howson et. al., 2023) does not require knowledge of the delay. We want to clarify that the mentioned paper considers an episodic setup where the rewards are non-composite and non-anonymous. So, their ideas may not directly translate to our infinite horizon setup with composite and partially anonymous rewards. Our algorithm requires knowledge of $d$ because it is a model-based setup where the confidence bounds must be accurately designed such that, after a sufficiently long horizon, the actual MDP falls in the confidence set with high probability. To our current understanding, $d$ plays a crucial role in defining the modified confidence bound that cannot simply be eliminated. In the future, if we look for an algorithm that does not require the knowledge of $d$, we must focus our attention on model-free procedures.
>
> $\textbf{Q3: On the related works section.}$
>
> $\textbf{Response:}$ In the revised manuscript, we have expanded the related works section. In particular, we have discussed works related to traditional RL setup.
>
> $\textbf{Q4: Discussion regarding partially anonymous feedback.}$
>
> $\textbf{Response:}$ In the modified paper, we provide the following example of an MDP with a partially anonymous reward (section 2).
>
> > Before concluding, we would like to provide an example of an MDP with partially anonymous rewards. Let us consider the $T$ round of interaction of a potential consumer with a website that advertises $S$ categories of products. At round $t$, the state, $s_t\in\\{1,\cdots, S\\}$ observed by the website is the category of product searched by the consumer. The $s$-th category where $s\in\\{1, \cdots, S\\}$ has $N_s$ number of potential advertisements, and the total number of advertisements is $N=\sum_{s}N_s$. The website, in response to the observed state, $s_t$, shows an ordered list of $K<N$ advertisements (denoted by $a_t$), some of which may not directly correspond to the searched category, $s_t$. This may cause the consumer, with some probability, to switch to a new state, $s_{t+1}$ in the next round. For example, if the consumer is searching for ''Computers'', then showing advertisements related to ''Mouse'' or ''Keyboard'' may incentivize the consumer to search for those categories of products later. After $\tau$  rounds, $0<\tau\leq T$, the consumer ends up spending $r_{\tau}(s)$ amount of money for the $s$-th category of product as a consequence of previous advertisements. Note that if the same state $s$ appears in two different rounds $t_1<t_2<\tau$, the website can potentially show two different ordered lists of advertisements, $a_{t_1}, a_{t_2}$ to the consumer. However, it is impossible to segregate the portions of the reward $r_{\tau}(s)$ contributed by each of those actions. Hence, the system described above can be modeled by an MDP with delayed, composite, and partially anonymous rewards.

---

> ### Author Response · Authors · 2023-06-28
> **Response: Part 2**
>
> $ \textbf{Q5: On the necessity of partially anonymous reward.}$
>
> $\textbf{Response:}$ Although it is difficult to assess whether partially anonymous feedback is a necessity for $\textit{any}$ algorithm, we can present a convincing argument explaining why it must be a requirement for all algorithms that need an accurate estimation of the reward function. Our algorithm falls into this category. In a  bandit setup with delayed, composite, and anonymous rewards, the estimation of $r(a)$, the reward associated with action, $a$ is obtained by contiguously executing $a$ and computing the average of the observed rewards. Clearly, if the number of times $a$ is executed is sufficiently large, the obtained estimate will be close to $r(a)$. However, in an RL setup, it is impossible to guarantee that a given state-action pair $(s, a)$ is contiguously visited. The most that one can ensure is that, within a given interval, the action, $a$ is executed whenever the state, $s$ is observed. Even with this strategy, it will be, in general, difficult to obtain an accurate estimate of $r(s, a)$ if one only has access to a fully anonymous reward (where the contributions arising from all previously visited state-action pairs are merged together). In contrast, if the reward is partially anonymous, then the learner gets access to the contribution arising from all previous visits only to the state, $s$, and it is not contaminated by the contributions arising from the previous visits to other states. In such a case, by executing the action, $a$ at state $s$ sufficiently large number of times, the learner can obtain an accurate estimate of $r(s, a)$. In conclusion, in order to get rid of the assumption of partial anonymity and deal with the most general setup of full anonymity, we have to design algorithms that do not require an accurate estimation of the reward function.
>
> $\textbf{Q6: Parsing the definition of $d$.}$
>
> $\textbf{Response:}$ We have added the following passage in the revised manuscript (Section 3) that elaborates the intuition behind assumption 2.
>
> > To better understand the intuition behind Assumption 2, consider an interval $\\{1, \cdots, T_1\\}$. Clearly, the reward sequence generated at  $t_1=T_1-\tau_1$, $\tau_1\in\\{0, \cdots, T_1-1\\}$, is $\mathbf{r}\_{t_1}(s_{t_1}, a_{t_1})$. The portion of this reward that is realized after $T_1$ is expressed by the following quantity: $\sum\_{\tau\geq \tau\_1}r_{t_1,\tau}(s\_{t_1}, a_{t_1})$ which is upper bounded by $\max\_{(s, a)} \sum\_{\tau\geq \tau_1} r_{t_1, \tau}(s, a)$. Therefore, the total amount of reward that is generated in $\\{1, \cdots, T_1\\}$ but realized after $T_1$ is bounded by $\sum\_{\tau_1\geq 0}\max_{(s, a)} \sum_{\tau\geq \tau_1} r_{t_1, \tau}(s, a)$. As the distribution of the reward sequence $\mathbf{r}_{t_1}$ is the same for all $t_1$ (Assumption 1(a)), one can replace the $\tau_1$ dependent term $t_1$ with a generic quantity, $t$. Thus, Assumption 2 essentially states that the $\textit{spillover}$ of the rewards generated within any finite interval $\\{1, \cdots, T_1\\}$ can be bounded by the term $d$.
>
> $\textbf{Q7: Precise definition of $T(s'|s, \pi, M)$.}$
>
> $\textbf{Response:}$ In the revised manuscript, we have provided a precise definition of the mentioned term.
>
> $\textbf{Q8: References before Appendix.}$
>
> $\textbf{Response:}$ We have made changes accordingly.
>
> $\textbf{Q9: Minor changes.}$
>
> $\textbf{Response:}$ We have gone through all the changes suggested by the reviewer and incorporated them into our revised manuscript. We thank the reviewer for the suggestions.

---

### Review · Reviewer_zAJN · 2023-06-18

**Summary Of Contributions:**

The paper consider infinite horizon average reward RL with delay, composite, and anonymous feedback. The authors propose the DUCRL2 algorithm to address this problem and analyze its regret bound.

**Audience:**

Yes

**Broader Impact Concerns:**

This paper is a theory paper and I am not immediately aware of any such concerns.

**Claims And Evidence:**

Yes

**Requested Changes:**

Following the above comments, the authors need to add the following elements:
1. Give a concrete and detailed example where this model could apply, but usual MDP (possibly with function approximation) cannot.
2. The paper needs to do a better job in stating the technical novelty. Otherwise, it reads like the technique is a simple combination of existing ones.

**Strengths And Weaknesses:**

Strengths:
1. The paper appears to consider a problem that was not previously studied in the RL literature;
2. The presentation of the model and the algorithm is crystal clear.

Weaknesses:
1. I am not entirely convinced by the motivation of the problem. Isn't the introduction of "states" exactly for the purpose of addressing long-term/delayed feedback? For example, in advertisement, the user may make additional clicks before purchasing anything. It appears to me that the additional clicks can be encapsulated by the "states". Delay and composite feedback would be a problem in bandit because bandit does not have states. But for MDP, I do not understand why a better characterization of the states cannot resolve this issue.
2. What is the technical novelty of the paper compared to existing works? Right now, the algorithm seems to be UCRL2 + bandit with delayed feedback.

---

> ### Author Response · Authors · 2023-06-29
> **Response: Part 1**
>
> $\textbf{Q1: Is delay captured by the states?}$
>
> $\textbf{Response:}$ In typical RL problems, the state is defined as an observable property of the environment that may change over time. On the other hand, a reward is a form of feedback generated by the environment when a certain action is executed at a certain state. Note that states may be of arbitrary type while rewards must be real numbers. In this paper, we assume that the learner does not get access to the reward immediately after executing an action. In other words, the delay is associated with the realization of the reward and, in general, it cannot be captured by the states. We would like to conclude with the comment that a few articles in the literature also analyze the effect of delayed reward in RL [1]-[2]. It corroborates the idea that state and delayed reward are two distinct entities and one cannot be replaced by the other.
>
> [1] Tal Lancewicki, Aviv Rosenberg, and Yishay Mansour. ''Learning adversarial Markov decision processes with delayed feedback''. In Proceedings of the AAAI Conference on Artificial Intelligence, volume 36, pages 7281–7289, 2022.
>
> [2] Benjamin Howson, Ciara Pike-Burke, and Sarah Filippi. ''Optimism and delays in episodic reinforcement learning''. In International Conference on Artificial Intelligence and Statistics, pages 6061–6094. PMLR, 2023.
>
> $\textbf{Q2: Concrete example.}$
>
> $\textbf{Response:}$ In the modified paper, we provide the following example of an MDP with a partially anonymous reward (section 2).
>
> > Before concluding, we would like to provide an example of an MDP with partially anonymous rewards. Let us consider the $T$ round of interaction of a potential consumer with a website that advertises $S$ categories of products. At round $t$, the state, $s_t\in\\{1,\cdots, S\\}$ observed by the website is the category of product searched by the consumer. The $s$-th category where $s\in\\{1, \cdots, S\\}$ has $N_s$ number of potential advertisements, and the total number of advertisements is $N=\sum_{s}N_s$. The website, in response to the observed state, $s_t$, shows an ordered list of $K<N$ advertisements (denoted by $a_t$), some of which may not directly correspond to the searched category, $s_t$. This may cause the consumer, with some probability, to switch to a new state, $s_{t+1}$ in the next round. For example, if the consumer is searching for ''Computers'', then showing advertisements related to ''Mouse'' or ''Keyboard'' may incentivize the consumer to search for those categories of products later. After $\tau$  rounds, $0<\tau\leq T$, the consumer ends up spending $r_{\tau}(s)$ amount of money for the $s$-th category of product as a consequence of previous advertisements. Note that if the same state $s$ appears in two different rounds $t_1<t_2<\tau$, the website can potentially show two different ordered lists of advertisements, $a_{t_1}, a_{t_2}$ to the consumer. However, it is impossible to segregate the portions of the reward $r_{\tau}(s)$ contributed by each of those actions. Hence, the system described above can be modeled by an MDP with delayed, composite, and partially anonymous rewards.

---

> ### Author Response · Authors · 2023-06-29
> **Response: Part 2**
>
> $\textbf{Q3: Technical Novelty.}$
>
> $\textbf{Response:}$ We have elaborated on our technical novelty in section 1.1 of the manuscript. Here we quote some relevant excerpts for easy reference. This explains why solving an MDP with delayed, composite, and anonymous reward is significantly more challenging than its bandit counterpart. Also, it remarks on why we are required to use the assumption of partial anonymity (instead of the more general assumption of full anonymity) to resolve the issue.
>
> > Learning with delayed, composite, and anonymous rewards has gained popularity in the RL community. While there has been extensive theoretical analysis of these types of rewards in the multi-arm bandit (MAB) framework, extending these ideas to the full Markov decision process (MDP) setting poses significant challenges. For example, one of the main ideas used in the MAB framework can be stated as follows (Wang et al., 2021). The learning algorithm proceeds in multiple epochs with the sizes of the epochs increasing exponentially. At the beginning of each epoch, an action is chosen and it is applied at all instances within that epoch. Due to multiple occurrences of the chosen action, and exponentially increasing epoch lengths, the learner progressively obtains better estimates of the associated rewards despite the composite and anonymous nature of the feedback. In an MDP setting, however, it is impossible to ensure that any state-action pair (s, a)
> appears contiguously over a given stretch of time. The most that one can hope to ensure is that each state, $s$ when it appears in an epoch (defined appropriately), is always paired with a unique action, a. In this way, if the state in consideration is visited sufficiently frequently in that epoch, the learner would obtain an accurate estimate of the reward associated with the pair (s, a). Unfortunately, in general, there is no way to ensure a high enough visitation frequency for all states. Unlike the MAB setup, it is, therefore, unclear how
> to develop regret optimal learning algorithms for MDPs with delayed, composite, and anonymous rewards.
>
> > Our article provides a partial resolution to this problem. In particular, we demonstrate that if the rewards are delayed, composite, and partially anonymous, then an algorithm can be designed to achieve near-optimal regret $\cdots$ A partially anonymous setup allows a learner to observe the sum of (delayed) reward components generated as a result of all past visitations to any specified state. Our proposed algorithm, DUCRL2, is built upon the UCRL2 algorithm of (Jaksch et al., 2010) and works in multiple epochs. Unlike the bandit setting, however, the epoch lengths are not guaranteed to be exponentially increasing. Our primary innovation lies in demonstrating how an accurate reward function estimate can be obtained using the partially anonymous feedback

---

### Decision · Action_Editors · 2023-08-22

**Recommendation:** Accept as is

**Comment:**

The reviewers all agree that the claim is well supported and the paper is of interest to some of the TMLR readers.

The main concern is about the assumptions made in the work, notably the partially anonymous reward feedback and the knowledge of delay. Some of the reviewer argues that with the current assumptions, the extension reduces the overall technical interest of the work. It is encouraged that the authors dedicate more discussions around this aspect.

Given the overall comments and recommendations from the reviewers, I am happy to support an accept decision.

**Audience:**

Yes.

**Claims And Evidence:**

The claims are supported by theoretical justifications, though some reviewers raised concerns about the novelty of the algorithm and necessity of the assumptions.

**Resubmission Of Major Revision:**

The authors may consider submitting a major revision at a later time.

---

> ### Author Response · Authors · 2023-08-28
> **Camera-Ready Submitted**
>
> Dear Editor,
>
> Thank you for smoothly conducting the review and accepting the paper. We have submitted the camera-ready version. Please let us know if anything else is needed from our side.